# The PROCAN-B study protocol: Early diagnosis of PROstate CANcer for Black men—a community-centred participatory approach in Scotland and the North-East of England

**Floor Christie-de Jong**[1]*, **Judith Eberhardt**[2], **Jonathan Ling**[3], **Marie Kotzur**[4], **Olugbenga Samuel Oyeniyi**[1], **Lawrence Nnyanzi**[5], **John Kabuye**[1], **Martin Kalemba**[1], **Kathryn A. Robb**[6]

1 Faculty of Health Sciences and Wellbeing, School of Medicine, University of Sunderland, Sunderland, United Kingdom, 2 Centre for Applied Psychological Science, Teesside University, Middlesbrough, United Kingdom, 3 Faculty of Health Sciences and Wellbeing, University of Sunderland, Sunderland, United Kingdom, 4 Dental School, University of Glasgow, Glasgow, United Kingdom, 5 Centre for Public Health, Teesside University, Middlesbrough, United Kingdom, 6 School of Health & Wellbeing, University of Glasgow, Glasgow, United Kingdom

* Floor.christie@sunderland.ac.uk

## Abstract

### Background

Prostate cancer is the most common cancer in the UK and Black African-Caribbean men are twice as likely to develop prostate cancer as white men. These cancer inequalities need urgent tackling. Barriers to early diagnosis are complex and require complex solutions. Culturally-tailored, community-centred and participatory approaches show promise in tackling cancer inequalities. We aim to co-design a culturally appropriate intervention to tackle barriers to early diagnosis of prostate cancer for Black men in Scotland and the North-East of England using a community-centred participatory approach.

### Methods

The PROCAN-B study is a mixed methods study set in Scotland and the North-East of England. A Public Involvement and Community Engagement (PICE) group (n = 12), is involved at every step of the research. Drawing on principles of the Integrated Screening Action model (I-SAM), the study has 8 objectives: 1) to explore barriers to early diagnosis of prostate cancer among Black men (45+) through focus groups (n = 12); 2) to co-design a culturally acceptable peer-led intervention to tackle barriers to early diagnosis of prostate cancer in Black men; 3) to train members of the community as 'peer-facilitators' (n = 8); 4) to deliver the intervention in each location, facilitated by peer-facilitators, with a purposive sample (n = 20) of Black men (45+); 5) to qualitatively evaluate the intervention through focus groups; 6) to refine the intervention based on qualitative feedback; 7) to pilot the refined intervention with another purposive sample (n = 40) through a cross-sectional survey pre-

**Data Availability Statement:** No datasets were generated or analysed during the current study. All

relevant data from this study will be made available upon study completion.

**Funding:** Authors who received this award are: FC, JL, JE, MKo, JK, KR. This study is funded by Prostate Cancer Research https://www.prostate-cancer-research.org.uk/ Grant reference number: 6968 The funders had no role in study design, data collection and analysis, decision to publish, or preparation of the manuscript.

**Competing interests:** The authors have declared that no competing interests exist.

and post-intervention; 8) to qualitatively evaluate the refined intervention through focus groups to further refine the intervention.

## Discussion

Community-centred and culturally tailored interventions have potential to be effective in addressing barriers to early diagnosis of prostate cancer, and thus ultimately reduce morbidity and mortality rates through earlier diagnosis in Black communities.

## Introduction

Prostate cancer is the most common cancer in men in Europe and the UK [1] with incidence and mortality rates projected to rise [2]. Black African and Caribbean men are at least twice as likely to develop prostate cancer as white men and twice as likely to die [3–7], presenting large cancer inequalities (n.b. African-Caribbean here refers to people of African ancestry with origins from the Caribbean, who may identify as mixed heritage, Black British, Black American, and so forth. For simplicity, we use the term 'Black', although we appreciate that this is a diverse group). Black men have been found to present at a younger age, with more aggressive disease and/or at later stages [8–10]. Recent evidence from the UK found that Black men are more frequently diagnosed with prostate cancer, although not necessarily at more advanced stages compared to White men, which is in contrast to US findings, although this was inconclusive in the younger age group (40–49) [5,11].

Although ethnic variations in prostate cancer outcomes are not yet fully understood [12], Black men understanding their risk of prostate cancer and recognising its symptoms could improve early diagnosis [13,14]. Yet, it appears that only 24% of Black men are aware of their higher risk of prostate cancer [15]. Although there currently is no national screening programme for prostate cancer in the UK yet, it is vital that Black men are aware of prostate cancer and their elevated risk as this can encourage timely help-seeking behaviour, and in turn early diagnosis. In the UK, all men over the age of 50 are entitled to ask for a PSA test, and Black men aged 45+ are encouraged by the NHS to talk about their risk with their doctor [16]. Early diagnosis of prostate cancer is important and will most likely mean the cancer is easier to treat and the higher the chance of successful treatment [3].

In addition to lack of awareness, barriers to early diagnosis of prostate cancer are complex and multi-factorial, ranging from lack of knowledge to social, economic, emotional, cultural and structural barriers [9,10,17–20]. For example, barriers such as communication and trust issues with healthcare providers, embarrassment, fears of the procedure and the outcome, or of being emasculated, have been described in the literature [9,10,13,14,17]. Trying to overcome such barriers to early diagnosis could encourage help-seeking behaviours in this population. There is limited research in the UK on understanding barriers to early diagnosis of prostate cancer for Black British African-Caribbean men [9,17]. In one study conducted in the UK exploring perspectives of a digital rectal examination (DRE) as a barrier to prostate cancer diagnosis for Black men, the fear of homophobia was found to be a major barrier [21]. The authors concluded that this fear must be addressed in collaboration with the community [21]. In addition, the multifactorial nature of these barriers, and corresponding facilitators, indicates that interventions aimed at increasing early diagnosis should be multidimensional. Complex public health issues such as improving early diagnosis of prostate cancer, require complex solutions, which is in line with the socio-ecological conceptual framework for public health, a

multi-level and interactive framework [22]. The framework is founded on the idea that in population health, individual health problems are multifactorial and cannot be explained or improved without examining multiple influences on health outcomes, including the wider social context in which individual health problems were created [22].

According to the socio-ecological framework, a public health issue like inequalities in prostate cancer is the result of a convergence of all factors involved. Therefore, it is unlikely that health education with the aim of raising awareness of prostate cancer risk alone is sufficient to tackle this complex issue and increase early diagnosis. Rather, multidimensional interventions that tackle multiple barriers, are more likely to be effective. It is important to understand how social structures impact on knowledge, practices, and barriers to early diagnosis of prostate cancer. Participatory and community-centred approaches are an important strategy to improve health and tackle health inequalities [23,24]. Interventions that address the expressed or identified needs of the target population by employing community engagement and that involve peers in intervention delivery, offer not only an ethical approach to health improvement, but have also been found to be effective in terms of changing and improving health behaviour, health consequences, participant self-efficacy and perceived social support for disadvantaged groups [24]. Although there is no clear understanding of the mechanisms through which community engagement approaches work, the evidence suggests that community engagement interventions have potential to achieve improvement in health outcomes and reduce health inequalities [24]. Furthermore, targeted and culturally tailored approaches in cancer communication that are aligned with the norms and values of the target population, are more effective than 'one size fits all' non-targeted interventions [25,26]. Therefore, working in partnership with Black populations ensures the intervention is culturally appropriate and therefore has the potential to be effective in addressing barriers to early diagnosis of prostate cancer, and thus ultimately reduce morbidity and mortality rates through earlier diagnosis in Black communities. Peer-led interventions can also help build trust among communities [25] and overcome barriers such as embarrassment and lack of trust [9,17]. There is some evidence that culturally appropriate and co-produced interventions are effective in raising awareness of the risk of prostate cancer and encouraging engagement with prostate cancer health checks among Black men in the US [27–29]. Community engagement models have been effectively used through multiple types of interventions, including faith-based approaches, peer-led educational sessions as well as technological tools such as apps [27–29]. As far as we are aware, only three UK studies have tried using a community engagement model to increase early diagnosis of prostate cancer among Black men [30–32]. One UK study co-designed in 2015 an interactive, educational and culturally appropriate game to raise awareness of prostate cancer among Black men. The game was qualitatively evaluated through focus groups (n = 29) and positively received [30]. It is unclear whether the game is currently in use. In 2020, UK researchers aimed to co-produce an app to raise awareness of the risk of prostate cancer among Black men [31]. Feedback on the app was mixed, with some men believing the app to reinforce racial stereotypes. The authors highlight the importance of ensuring a diverse sample engaged in the co-production, as well as collecting early feedback. The third UK study co-created a video to raise awareness of prostate cancer risk with seven Black men. The video was not formally evaluated [32]. More work is needed in the UK to encourage in a culturally appropriate manner early diagnosis of prostate cancer among Black British African-Caribbean men. Therefore, in this study we aim to co-design a culturally appropriate intervention to tackle barriers to early diagnosis of prostate cancer for Black men in Scotland and the North-East of England using a community-centred participatory approach.

## Methods

### Theoretical framework

This research will draw on the principles of the Integrated Screening Action Model (I-SAM) (Fig 1) [33]. The I-SAM proposes an integrated model to understand cancer screening behaviour and serves as a practical tool to design interventions aimed at improving screening uptake by identifying potential targets and policies to increase access to screening. The I-SAM synthesises existing models of health behaviour which includes three main aspects: 1) the I-SAM outlines a sequence of stages individuals go through when engaging in precautionary behaviour, based on the stages of behaviour change of the Precaution Adoption Process Model [34], 2) the I-SAM acknowledges that cancer protective behaviour is shaped and influenced by the interrelationships between individual, social and environmental factors of the socio-ecological model and consequently distinguishes between participant and environmental influences [33], and 3) the I-SAM highlights the importance of targeting sources of behaviour such as capability, opportunity, and motivation targets for behaviour change of the COM-B model, which incorporates the Behaviour Change Wheel and outlines nine intervention functions [35,36]. The I-SAM conceives capability and motivation as participant influences, and opportunity as environmental influences. For example, capability includes (but is not limited to) constructs such as knowledge, self-efficacy or transport; motivation includes emotions and perceived risk. Opportunity includes constructs such as social norms and stigma, and physical opportunity could entail patient navigation and ease of accessing healthcare (Fig 1). The I-SAM supports our understanding of health behaviour in relation to cancer and will underpin the entire study, including data collection, data analysis and intervention development as it provides clear guidance on how to conceptualise constructs and translate these into intervention functions and policies.

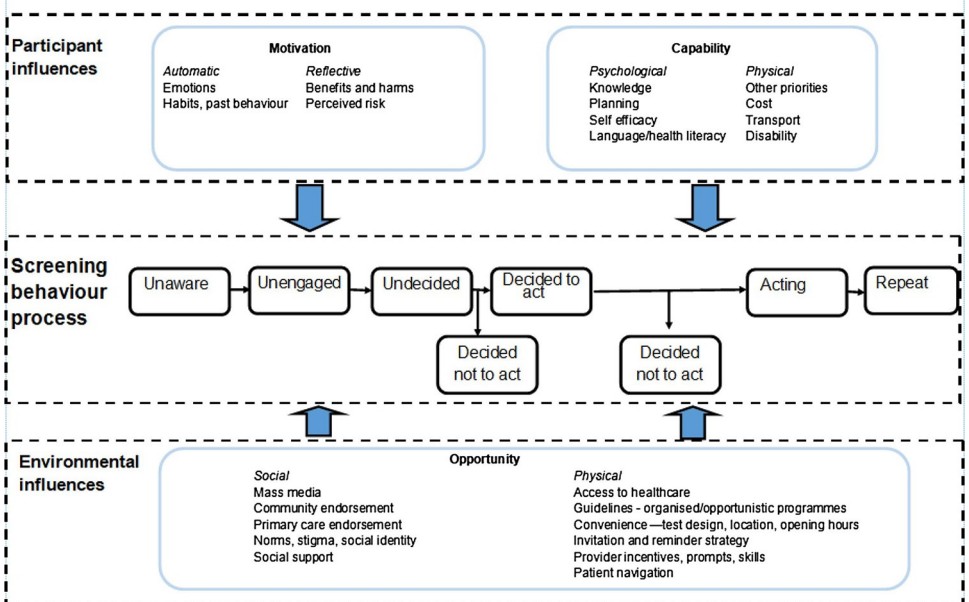

**Fig 1. The Integrated Screening Action Model (I-SAM) Reproduced from Robb KA.** The integrated screening action model (I-SAM): A theory-based approach to inform intervention development. Prev Med Rep. 2021 Sep 1;23:101427 under the Creative Commons CC BY 4.0 license (https://creativecommons.org/licenses/by/4.0/). Copyright © [2021] Elsevier [33].

Table 1. Study objectives.

| Work Package | Objective |
|---|---|
| WP1 | To explore barriers to early diagnosis of prostate cancer among Black men in Scotland and the North-East of England. |
| WP2 | To co-design a culturally acceptable peer-led intervention to tackle barriers to early diagnosis of prostate cancer in Black men. |
| WP3 | To train members of the PICE group as 'peer-facilitators'. |
| WP4 | To deliver the intervention, facilitated by peer-facilitators, to Black men aged 45+. |
| WP5 | To qualitatively evaluate the intervention through focus groups |
| WP6 | To refine the intervention after evaluation. |
| WP7 | To pilot the refined intervention through a cross-sectional survey pre- and post-intervention. |
| WP8 | To qualitatively evaluate the refined intervention |

## Research design

This is a mixed methods study design with qualitative and quantitative components. The mixed methods design is an exploratory sequential design which begins with an exploratory phase prioritising the collection of qualitative data. The qualitative phase will inform the intervention design. Building from the exploratory results, and based on the I-SAM, quantitative data will be collected to allow for pilot testing of intervention effectiveness [37].

The study takes a community-centred and participatory approach, working in partnership with Black communities. The study will run in two socioeconomically deprived areas with poor cancer outcomes: Scotland and the North-East of England, which lack inclusion in UK research. Including Black men from less ethnically diverse settings, such as Scotland and the North-East of England, is important to ensure their voices are heard. These two settings also incorporate different healthcare systems, which could be important for future feasibility testing, and are pragmatic choices as they are where the researchers are based.

The study has eight objectives, which are aligned to the study's work packages (Table 1).

## Sample, sampling and recruitment

The target population for the study participants is any Black man aged 45+ living in the North East of England or Scotland, without a clinical diagnosis of prostate cancer, who meets the criteria below:

Inclusion criteria:

- Identifying as a Black male, age 45 years and above and living in either North-East of England or Scotland.

Exclusion criteria:

- Participants who do not identify as Black male, age 45 years and above, or living either in North-East of England or Scotland.

- Men who have had prostate cancer before, or have had a prostatectomy for other reasons, will not be eligible to participate as intervention participants, as their knowledge of prostate cancer and attitudes to help-seeking will likely differ from men who have not had prostate cancer and/or a prostatectomy.

To obtain wide-ranging perspectives and ensure a diverse sample, we will aim to use purposive sampling to target participants (aged 45+) of various age groups, as well as different socio-economic backgrounds and ethnicities. For example, including a diverse sample of men born

**Table 2. Sample size specified per objective.**

| Study Phase | Sample |
|---|---|
| Objective 1-exploration of barriers | PICE group N = 12 |
| Objective 2-design of intervention | PICE group N = 12 |
| Objective 3-training of peer facilitators | PICE group N = 8 |
| Objective 4-delivery of intervention | Intervention 1 participants:<br>N = 20 (n = 10 in North-East, n = 10 in Scotland) |
| Objective 5-evaluation of intervention | Intervention 1 participants: N = 20 (n = 10 in North-East, n = 10 in Scotland) |
| Objective 6-refinement of intervention | PICE group N = 12 |
| Objective 7-pilot test intervention | Intervention 2 participants:<br>N = 40 (n = 20 in North-East, n = 20 in Scotland) |
| Objective 8-evaluate intervention | From Intervention 2 participants: N = 20 (n = 10 in North-East, n = 10 in Scotland) |
| **Total overall** | **PICE group N = 12**<br>**Study participants N = 60** |

in the UK and/or of African origin from various countries across the African continent, as well as from the Caribbean. Snowball sampling will also be applied to ensure sample sizes are achieved. Table 2 specifies the sample and sample size per objective.

Participants in the first delivery (n = 20) and the second delivery (n = 40) of the intervention are different men with a total target sample size of 60. As this is a pilot study, formal sample size calculations are not required [38]. In a review of 761 studies, the median target sample size was 30, with a range between 20 and 50 [39]. Our target sample size of 60 seems therefore appropriate, with a smaller group in the first delivery of the intervention to allow feedback on the second delivery of the refined intervention and the pre-post survey testing with a slightly larger group.

The recruitment process will be led by JK, a collaborator on the project and employed as the part-time Recruitment Lead for the North-East of England and MKa, the part-time Recruitment Lead for Scotland. JK and MKa are members of relevant communities and are an important part of the research team. They are also members of two community organisations that have agreed to collaborate with the study. JK and MKa will recruit through word of mouth and through their Black-owned/Black dominated community organisations, which will start in November 2023. The advertisement for the study will be communicated through these community organisations, such as posted on the organisations' social media, websites, email, messenger or SMS where possible. If recruitment takes place through other community organisations which require involvement of gatekeepers, gatekeepers can explain the study to potential participants but will not be actively or directly recruiting. Gatekeepers can share information about the study, such as the participant information sheet, with interested individuals. Interested individuals may contact the research team directly, or the gatekeeper who will pass their contact information to the research team. Information about the study, such as the participant information sheet, will be provided to each participant in writing personally, via email, or other virtual means, such as messenger, depending on participant's preference. The research team will contact the participant directly by phone or email to review the participant's information and address any questions. If the person agrees to participate, an invite to the study session(s) can be sent.

## Ethical considerations

Ethical approval for all work packages was obtained from the Research Ethics Committee of the University of Sunderland on the 7th of December 2022 (#015660). We will comply with the UK Research Integrity Office *Code of Practice for Research* throughout the project. Non-coercive recruitment methods will be used such as posters and flyers. Participants will be reimbursed for travel and expense cost based on £25 per hour. Informed consent will start with offering participants the participant information sheet, which will cover in clear and accessible language what is involved in participating; benefits and risks; terms for withdrawal; usage of the data; strategies for assuring ethical use of the data; contact details; and how to file a complaint. Ahead of providing written consent, participants will be given the time to consider the study information and will be given the opportunity to ask questions about the nature and objectives of the study and possible risks associated with their participation. To make the Participant Information Sheet as accessible as possible, summarised information will be presented in video format too. Participant names, telephone numbers and/or email addresses will be collected by the Recruitment Leads. These details will be kept in a separate file and transferred to OneDrive from Outlook or research team members' mobile phones; removal of the data from the email/phones will be made as a precaution after the transfer to OneDrive. Access to files in OneDrive will be restricted to researchers from the project's research team only. We will also ensure that documents are not downloadable onto other devices. After consent, each participant will be allocated a unique identification number. Audio recordings of interviews will be identified by this identification number and not by personal identifier. All anonymised data will be kept on the university's secure and password-protected OneDrive. Audio recordings will be transcribed by an external provider and any potentially identifying details will be removed by the research team.

## Work packages

The study comprises of eight work packages. The study adopts a progressive design to achieve its aim, to co-design a culturally appropriate intervention to tackle barriers to early diagnosis of prostate cancer for Black men in Scotland and the North-East of England using a community-centred participatory approach. Each work package builds upon findings from the previous work package. This iterative approach ensures a comprehensive and coherent intervention development process. As each of the work packages has its distinct objectives and methods, we present the work packages separately.

**Work Package 1 –exploring barriers to early diagnosis of prostate cancer.** The objective of Work Package (WP)1 is to explore barriers to early diagnosis of prostate cancer among Black men in Scotland and the North-East of England. We will recruit a Public Involvement and Community Engagement (PICE) group (n = 12) of men from Black communities combined from the North-East of England and Scotland. Based on existing qualitative research, a sample size of 12 is likely sufficient for this qualitative component, as it allows for in-depth exploration of themes, potentially reaching data saturation [40]. The Recruitment Leads will recruit the PICE group through the community organisations. To obtain wide-ranging perspectives, we use purposive sampling to target participants (aged 45+) and aim for diversity regarding characteristics such as a) age, b) ethnicity, c) location. We will explore barriers and facilitators to early diagnosis of prostate cancer to Black men in two online focus groups with PICE members. This qualitative participatory approach allows for wide-ranging perspectives to be incorporated into the intervention design [41]. To ensure data saturation can be reached, we employ some flexibility in our data collection process and may conduct a third focus group to reach data saturation, which could allow depth in the data and a more in-depth

understanding of the issues involved. The focus groups will be digitally audio recorded and transcription outsourced. The outcome of WP1 is to identify key barriers and facilitators to early diagnosis of prostate cancer, to underpin intervention development.

**Work Package 2 –prototype intervention development.** The objective of WP2 is to co-design a culturally acceptable peer-led intervention to tackle barriers to early diagnosis of prostate cancer in Black men. Findings from the initial focus groups in WP1 provide a foundation for the iterative process of intervention development through continued discussion in three co-design workshops. This co-design phase uses a participatory approach with three online workshops with the PICE group based on the World Café method to develop the intervention [42]. The World Café method is a research method that focuses on fostering collaborative dialogue and conversation between participants in a group setting and is based on providing a comfortable informal setting which facilitates the inclusion and exchange of diverse views on a topic [42]. This method enables researchers to engage more deeply with the communities they study and thus generates richer insights and is particularly useful for exploring topics and generating creative solutions to challenging problems [43]. The World Café method has been used successfully with marginalised groups and provides a structured yet flexible way to engage participants in meaningful dialogue and generate valuable insights and ideas. Participants are set up in small groups, conducive to free flow of conversation. Participants are then presented with the topic or question and asked to reflect on this for a few minutes, followed by a first round of discussions to share ideas and insights. Groups are then changed, and participants continue the conversation with a different group. This process is repeated several times, with the aim of conversations developing progressively and reaching deeper insights. At the end, participants come back together as a whole group to share insights and ideas emerged from the discussions. The workshops will be digitally audio-recorded and transcription will be outsourced. Thematic analysis will be used to analyse and map data from the WP1 focus groups and WP2 co-design workshops, to the I-SAM. Data analysis will be continuous and will feed into the next workshop. Themes, barriers and facilitators emerging from data collected in WP1, will be discussed with the PICE group in WP2 to find ways of developing potential solutions and addressing these barriers in the intervention. Based on ideas generated by the PICE group and an exploration of their views on barriers and ways of tackling these, a draft intervention strategy will be developed by the research team from the I-SAM's target intervention functions, and policy categories, to improve access to screening. The three co-design workshops will follow an iterative process, where the draft intervention strategies will be presented to the PICE group and feedback on intervention strategies will be collected and incorporated into the evolving draft design.

The intervention will be a peer-led, multidimensional community intervention. It will incorporate multiple components that tackle barriers to early diagnosis: such as health education, or possibly personal testimonials through survivors' stories, if the PICE group believe these are helpful. These components will be delivered in culturally appropriate ways. However, this initial prototype of the intervention will be informed by further discussions with the PICE group. APEASE criteria (Affordability, Practicability, Effectiveness and cost-effectiveness, Acceptability, Side-effects/safety, Equity) will be applied when deciding on the intervention strategy [44]. When deciding on an intervention strategy, each of these criteria should be considered and weighed against one another to determine the most appropriate course of action. By considering each of these criteria, it can be ensured that the intervention strategy is both effective and equitable, while also taking into account practical considerations such as cost and feasibility. The outcome of WP2 is the design of the intervention.

**Work Package 3 –peer facilitator training.** The objective of WP3 is to train members of the PICE group as 'peer-facilitators'. Peer-facilitators are important in prostate cancer

communication [45,46] and can act as trusted sources in intervention delivery. Peer-facilitators also have a vital role in the sustainability of community interventions. We will recruit the peer-facilitators from the PICE group and aim to have four peer-facilitators in each location. We aim to run three two-hour online workshops to train the peer-facilitators, which will provide sufficient time to enhance their confidence and ensure familiarity with their roles. A training plan will be developed that covers the essential topics and skills that the peer facilitators will need to be effective, which will include both theoretical and practical training, tailored to the specific needs and context of the community. We will use interactive training methods to engage the peer-facilitators such as role-playing, group discussions and hands-on activities. We will also provide on-going support to peer-facilitators including regular feedback and opportunities for continued development. After each session, we will collect feedback to ensure areas for improvement can be identified. The outcome of WP4 is to develop trained peer-facilitators to deliver the intervention.

**Work Package 4 –prototype intervention delivery.** The objective of WP4 is to deliver the intervention in December 2023, facilitated by the peer-facilitators, to Black men aged 45+. A purposive sample of Black men aged 45+ in Scotland (n = 10) and the North-East of England (n = 10) will be recruited by the recruitment leads through community organisations. The intervention will be delivered face-to-face to ensure inclusivity, although this does depend on decisions made by the PICE group and any COVID-19 restrictions. The intervention will be delivered once in each location. The outcome of WP4 is the initial delivery of the intervention.

**Work Package 5 –prototype intervention evaluation.** The objective of WP5 is to qualitatively evaluate the intervention through focus groups in December 2023. The initial intervention will be evaluated through two focus groups with the intervention participants (n = 20): one focus group in Scotland (n = 10) and one in the North-East of England (n = 10). We will aim to conduct the focus groups face-to-face. The focus groups will cover feasibility and acceptability of the intervention and the I-SAM will underpin the thematic analysis. Focus groups are well-suited to understanding how people think and talk about early diagnosis of cancer [47]. They present a more formalised approach to gathering design feedback than the co-design workshops. We will develop a topic guide to explore in the focus groups what men thought of the intervention's acceptability, its content, delivery, recommendations for improvements, and any potential impact they perceived the intervention to have in terms of knowledge and attitudinal change towards early diagnosis of prostate cancer. Acceptability of the revised intervention and materials will be explored using the Theoretical Framework of Acceptability (TFA). TFA defines acceptability as "*a multi-faceted construct that reflects the extent to which people delivering or receiving a healthcare intervention consider it to be appropriate, based on anticipated or experienced cognitive and emotional responses to the intervention*" [48]. TFA focuses on the extent to which an intervention is perceived as appropriate, relevant, and feasible by those who are expected to use it. It provides insights into the factors that influence the acceptability of an intervention, such as the perceived benefits and risks, complexity, compatibility with existing practices, and perceived social norms.

The outcome of WP5 provides insights and knowledge gained from the qualitative evaluation, which will inform further changes to the intervention.

**Work Package 6 –intervention refinement.** The objective of WP6 is to refine the intervention after the initial qualitative evaluation, early 2024. We will conduct an online focus group with the PICE group (n = 12) to review the thematically analysed qualitative findings from the focus groups with intervention participants in WP5. The online focus group data will be analysed using inductive thematic analysis. Based on the participants' perspectives on the intervention, and underpinned by the I-SAM model, the intervention content and delivery will be further adapted and refined in collaboration with the co-design group. The outcome of

WP6 will provide a refined intervention based on understanding what worked and what requires changing from the intervention's initial delivery.

**Work Package 7—refined intervention testing.** The objective of WP7 is to deliver the refined intervention and to pilot test the refined intervention through a cross-sectional survey pre- and post-intervention in 2024. This phase of the study utilises a quasi-experimental pre-post-test design. An experimental approach is not feasible within the timeline of the project. A purposive sample of Black men aged 45+ in Scotland (n = 20) and the North-East of England (n = 20) will be recruited through community organisations to receive the refined intervention. The I-SAM will inform the data collection tool. A Knowledge, Attitudes and Practice (KAP) survey will be constructed by adapting questions from the Knowledge of Prostate Cancer Questionnaire [49], Thomas Jefferson University Prostate Cancer Screening Survey [50], Precaution Adoption Process Model (PAPM) [51], Cancer Awareness Measure 2020 [52], and in line with the I-SAM a brief measure of constructs relating to capabilities, opportunities, and motivations [53], to assess help-seeking in relation to prostate cancer. The survey will be conducted prior to and post-exposure to the intervention. Surveys will be conducted using Computer-Assisted Telephone Interviewing (CATI). Telephone surveys tend to have higher response rates compared to self-completion surveys [54]. Two research assistants will be trained to conduct the telephone surveys 1–2 weeks before the intervention, and 2-weeks post-intervention. Qualtrics will be used to administer the survey online. Knowledge gained from this phase will inform future feasibility testing of the intervention and provide preliminary understanding of the intervention's impact.

**Work Package 8 –refined intervention evaluation.** The objective of WP8 is to qualitatively evaluate the refined intervention after the second delivery has been conducted in WP7. This qualitative evaluation will take a similar approach to the evaluation in WP5 and will again be underpinned by TFA. The refined intervention will be evaluated through two focus groups, using a similar topic guide used in WP5, unless change is required. A different group of will take part than in WP5, i.e. a selection of the WP7 refined intervention participants with one focus group in Scotland (n = 10) and one in the North-East of England (n = 10). After the intervention delivery, WP7 participants will be asked if they would be willing to take part in a focus group. This work package will help to achieve an in-depth understanding of participants' perspectives in the refined intervention. This will help to develop further refinement and improvements in the intervention strategies, which is key for the large-scale implementation of the intervention. This work package will help to understand what men thought of the refined intervention, its content, delivery, recommendations for improvements, and any potential impact they perceived the intervention to have in terms of knowledge and attitudinal change. The knowledge gained from the qualitative evaluation will inform further changes to the intervention. Knowledge gained from this phase will inform further changes to the intervention, which is key considering further testing of the intervention and future implementation.

The summary of the work packages and objectives is shown in the study's flow chart diagram below (Fig 2).

**Public involvement and community engagement.** Public Involvement and Community Engagement (PICE) is a crucial part of this project. Our participatory, community-centred co-design approach to the development of the intervention, is also entirely based on engagement with the community. The approach highlights our appreciation of ensuring the voice of the communities whose health we aim to improve, is included in the intervention design and development, to ensure public health interventions are culturally appropriate and effective. We therefore aim to work in close partnership with the Black community. Our PICE group informs the project at every stage. Having members of the community employed by the project

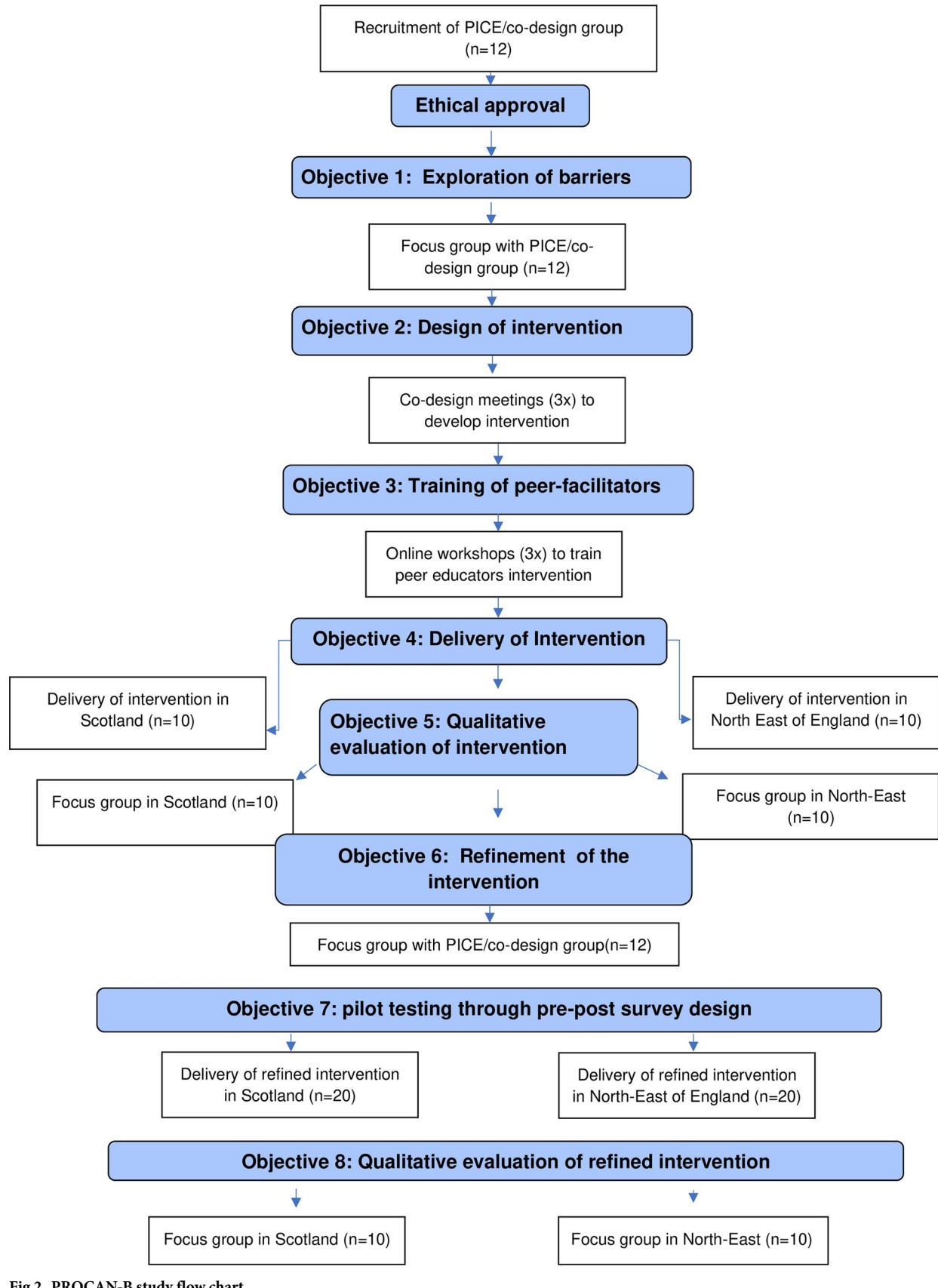

**Fig 2. PROCAN-B study flow chart.**

(i.e. the Recruitment Leads) and be part of the research team, and working closely with community organisations, makes this work truly participatory. We will present interim and final findings to the PICE group as the project continues, and seek their feedback throughout every stage of the project. Black men affected by prostate cancer are crucial to the success of our project. Including Black men who are prostate cancer survivors or who have engaged in help-seeking behaviours regarding prostate cancer, would be ideal as part of the PICE/co-design or peer facilitator group. It may be useful to include their stories in the intervention as personal testimonies, provided the co-design group believes this to be beneficial.

**Data collection.** The study comprises of multiple data collection points and methods, as specified in the description of the work packages. The qualitative focus groups and intervention sessions will be delivered by Black male members of the research team, if the PICE group believe this is preferred. The qualitative components in WP5 and WP8 will aim to conduct focus groups although individual interviews may be conducted if preferred by participants. In the pre-post-test design in WP7 telephone surveys will be administered in Qualtrics. Instruments required for the different phases of data collection are specified in Table 3.

**Table 3. Data collection instruments.**

| Data Collection Work package | Instruments |
|---|---|
| **Qualitative focus groups WP1** | A topic guide has been developed to explore barriers and facilitators to early diagnosis of prostate cancer (Box 1). Topics include (but are not limited to) barriers/enables to seeking help, perspectives regarding known barriers such as awareness, challenges to accessing healthcare, experiences with health care providers, embarrassment, cultural barriers, social stigma, and masculinity. |
| **Qualitative focus groups WP5** | A topic guide will be developed to explore feasibility and acceptability of the intervention. We will explore what men thought of the intervention, its delivery, recommendations for improvements, and any potential impact they perceived the intervention to have in terms of knowledge and attitudinal change. |
| **Pilot test WP7** | A Knowledge, Attitudes and Practice (KAP) survey has been constructed by adapting questions from the Knowledge of Prostate Cancer Questionnaire [49], Thomas Jefferson University Prostate Cancer Screening Survey [50], Precaution Adoption Process Model (PAPM) [51], Cancer Awareness Measure 2020 [52] |
| **Qualitative focus groups WP8** | A topic guide will be developed to explore feasibility and acceptability of the refined intervention. We will explore what men thought of the intervention, its delivery, recommendations for improvements, and any potential impact they perceived the intervention to have in terms of knowledge and attitudinal change. |

Box 1. Topic guide subjects and example questions

• **Knowledge and attitudes to prostate cancer** (e.g. What do you think or feel when you hear 'prostate cancer'? How do you think other Black men think/feel about prostate cancer?)

• **Sources of Information** (e.g. Where do you go for information about your health? (probe: if participants do not seek health information, probe to explore that; are there any differences in health seeking?)

• **Help seeking attitudes** (e.g. How do Black men feel about accessing healthcare?)

• **Healthcare providers** (e.g. How do you feel about your GP or nurse? (Probe: Do you feel comfortable with them? Do you trust them?)

• **Social and cultural factors** (e.g. How do you think discussing issues such as prostate cancer is perceived in the Black community? (probe: is this difficult? If so, why? How could change be achieved if change is needed?).

**Data analysis.** Data analysis occurs at multiple points, as specified below.

**Data analysis qualitative focus groups WP1:** All focus groups are audio recorded, and transcribed verbatim. Anonymised transcripts are thematically analysed using NVivo software to support data management and map data from the focus groups and the co-design workshops to the I-SAM. Thematic analysis is a widely used method for analysing qualitative data. It involves identifying patterns or themes within data that relate to the research question or objectives [55]. The analysis occurs in two stages. First, we inductively examine the data to identify barriers and facilitators to help-seeking drawing on the reflexive thematic analysis approach. This iterative method involves a number of stages including: familiarisation with the data; data coding; generation of initial codes, searching for themes; and reviewing of themes [55]. This approach relies on the researcher to engage thoughtfully with the data, systematically identify and label features in the data that seem important in relation to the research question and work collaboratively with other researchers to build a rich interpretation of the data. Next, we conduct more deductive analysis of data, guided by I-SAM constructs. This approach will help avoid 'forcing' data into pre-prescribed constructs and allow capture of unexpected/divergent issues. Each transcript is independently coded then discussed by two researchers. Themes generated by the analysis, and any disagreements between the two researchers, are discussed and agreed amongst the project team, with additional sense-making sessions involving the PICE group to ensure our findings reflect lived experience. A draft intervention strategy is developed from the I-SAM and the Behaviour Change Wheel's hubs of sources of behaviour (capability, opportunity and motivation) intervention functions, and policy categories. Intervention content and mode of delivery are based on the I-SAM's behaviour change strategy. Data analysis is continuous and feeds into the next workshop.

**Data analysis qualitative focus groups WP5 & WP8:** All focus groups and/or interviews will be audio recorded, and transcribed verbatim. Anonymised transcripts will be thematically analysed using NVivo software to support data management. Each transcript will be independently coded then discussed by two researchers. Themes generated by the analysis will be discussed and agreed amongst the project team, with additional sense-making sessions involving PICE groups to ensure our findings reflect lived experience.

**Data analysis Pilot test WP7:** We will collect data at two points in time–pre-intervention, and post-intervention, to be able to assess changes over time through visual inspection of the data. Visual inspection of data involves examining the data using graphs and charts to identify patterns, trends, outliers, and other features that can provide insight into the underlying phenomena being studied. Change scores will be calculated for each participant on each variable. Pre-and post-intervention scores will be plotted on a graph to assess trends in the sample. This will provide an indication of the feasibility of collecting these data in this setting, some indication of the likely effects of the intervention, and will inform a larger feasibility study based on our findings. The sample size (n = 40) of the pre-post survey design phase of the study is small and statistical analysis of the data may be of limited use, as there could be insufficient power, which can limit the ability to draw robust conclusions using traditional statistical methods [56]. However, to inform feasibility of the pre-post test design and measurement of independent and dependent variables, a McNemar's test to compare categorical data will be conducted to explore whether there is a change in the dependent variable 'intention to engage in prostate

cancer screening'. In line with mixed-methods designs, qualitative themes will be integrated with the quantitative findings which could be presented in a joined display [37].

## Discussion

Prostate cancer presents a significant health inequality for Black men. Black men have a higher mortality rate than any other racial or ethnic group [57]. Although ethnic variations in cancer outcomes are not yet fully understood [10], it is evident that cancer inequalities need to be addressed. Community-centred approaches and culturally tailored responses, such as the approach proposed in the PROCAN-B study, can help address cancer inequalities [58,59]. Early diagnosis is an essential component in reducing the impact of cancer on individuals and communities, is critical for improving the chances of survival and reducing the severity of prostate cancer. However, Black men are not sufficiently aware of the risks and symptoms of prostate cancer and they encounter barriers to help-seeking [13]. A community-centred and participatory approach can help raise awareness of the risk of prostate cancer and encourage Black men to seek early diagnosis. The history of systemic racism and discrimination in healthcare has led to a lack of trust in the medical system by many Black men [9]. This distrust can make it challenging for Black men to seek medical care, including prostate cancer detection. The strength of this work lies in the community-centred or participatory approach, which can help build trust by engaging Black men in the process of addressing the issues and empowering them to take charge of their health. There is a gap in knowledge regarding this important topic in the UK. Novel approaches to engaging with target populations in tailored ways, such as the PROCAN-B study, are critical to improving health outcomes for communities, particularly for the Black community, who are at increased risk of prostate cancer. Working in community and health partnerships would allow sustainable implementation of such health promotion efforts [58,59]. To maximise impact, study findings will be disseminated in multiple ways to reach a diverse audience, including the Black community, members of the public, public health professionals and practitioners, and academic audiences. Dissemination strategies will be discussed with the PICE group and will include dissemination events, accessible materials such as infographics and animation, as well as a publicly available manual detailing every step of the study. The manual could function as a toolkit for practitioners and academics to support participatory work. Study findings will also be written up in academic papers and presented at conferences. Although there will be limitations to drawing conclusions regarding the effectiveness of the intervention due to small sample sizes, the study does position itself well for a feasibility trial to explore whether effectiveness can be investigated more robustly. The study findings will not be generalisable due to the research design and small sample sizes, however detailed descriptions of the settings, participants, and methods of recruitment will be offered and findings could therefore be transferable to other settings. In particular, the participatory and community-centred approach will be transferable to other settings, populations, and public health issues. Future research could investigate the co-designed intervention more robustly to test effectiveness in multiple sites across the UK, with a representative sample to obtain generalisability. In conclusion, the community-centred and participatory approach applied in the PROCAN-B study to encourage early detection and diagnosis of prostate cancer for Black men is crucial for addressing health inequalities, increasing awareness, building trust, and improving outcomes.

## Author Contributions

**Conceptualization:** Floor Christie-de Jong, Judith Eberhardt, Jonathan Ling, Marie Kotzur, Kathryn A. Robb.

**Funding acquisition:** Floor Christie-de Jong, Judith Eberhardt, Jonathan Ling, Marie Kotzur, John Kabuye, Kathryn A. Robb.

**Investigation:** Floor Christie-de Jong, Olugbenga Samuel Oyeniyi, Lawrence Nnyanzi, John Kabuye, Martin Kalemba.

**Methodology:** Floor Christie-de Jong, Judith Eberhardt, Jonathan Ling, Marie Kotzur, Lawrence Nnyanzi, Kathryn A. Robb.

**Project administration:** Olugbenga Samuel Oyeniyi.

**Resources:** Floor Christie-de Jong, Judith Eberhardt, Olugbenga Samuel Oyeniyi, Lawrence Nnyanzi, John Kabuye, Martin Kalemba.

**Supervision:** Floor Christie-de Jong.

**Writing – original draft:** Floor Christie-de Jong.

**Writing – review & editing:** Floor Christie-de Jong, Judith Eberhardt, Jonathan Ling, Marie Kotzur, Olugbenga Samuel Oyeniyi, Lawrence Nnyanzi, John Kabuye, Martin Kalemba, Kathryn A. Robb.

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
