## [Decision Letter · Decision Letter 0]

12 Aug 2024

PONE-D-23-32907The PROCAN-B Study Protocol: Early diagnosis of PROstate CANcer for Black men - a community-centred participatory approach in Scotland and the North-East of EnglandPLOS ONE

Dear Dr. Christie-de Jong,

Thank you for submitting your manuscript to PLOS ONE. After careful consideration, we feel that it has merit but does not fully meet PLOS ONE’s publication criteria as it currently stands. Therefore, we invite you to submit a revised version of the manuscript that addresses the points raised during the review process.

We look forward to receiving your revised manuscript.

Kind regards,

Mohammed Faruk

Academic Editor

PLOS ONE

Journal Requirements:

"Authors who received this award are: FC, JL, JE, MKo, JK, KR. This study is funded by Prostate Cancer Research https://www.prostate-cancer-research.org.uk/ 

Grant reference number: 6968"

3. Please include a copy of Table 3 which you refer to in your text on page 17.

Reviewers' comments:

Reviewer's Responses to Questions

**Comments to the Author**

1. Does the manuscript provide a valid rationale for the proposed study, with clearly identified and justified research questions?

Reviewer #1: Yes

Reviewer #2: Yes

2. Is the protocol technically sound and planned in a manner that will lead to a meaningful outcome and allow testing the stated hypotheses?

Reviewer #1: Yes

Reviewer #2: Partly

3. Is the methodology feasible and described in sufficient detail to allow the work to be replicable?

Reviewer #1: Yes

Reviewer #2: No

4. Have the authors described where all data underlying the findings will be made available when the study is complete?

Reviewer #1: Yes

Reviewer #2: No

5. Is the manuscript presented in an intelligible fashion and written in standard English?

Reviewer #1: Yes

Reviewer #2: Yes

6. Review Comments to the Author

You may also provide optional suggestions and comments to authors that they might find helpful in planning their study.

Reviewer #1: Manuscript Feedback

Title: The PROCAN-B Study Protocol: Early diagnosis of PROstate CANcer for Black men - a

community-centred participatory approach in Scotland and the North-East of England.

Strengths.

This manuscript seeks to address a significant public health issue as prostate cancer is a leading

cause of death among men in the UK and disproportionately affects Black men.

This study seeks to design an intervention that is tailored to the needs and sensitivity of the

communities most impacted.

An attempt is made to engage the targeted communities for input in the design of the intervention.

Opportunities.

Framework.

A theoretical framework is provided: Integrated Screening Action Model. However, how this

framework informs the study design or other aspects of the study (Focus Group guide, refining

intervention testing, etc.) is not mentioned.

Sample Size/Recruitment.

Objective 1 – exploration of barriers. Sample group of N=13 for both sites is too small to give

diverse views from both communities.

Objective 2 – design of intervention. The sample size of 13 for both sites seem small to capture

diverse views, especially given the aim of designing an intervention that is culturally relevant to

each site.

Objective 6 – Same observation as for Objectives 1 &2 above

Objectives 4, 5, 7, 8, the sample size appear more reasonable.

Work Package 2.

Are the participants involved in this Work Package, the same 13 participants recruited for Work

Package 1? If so, how would you ensure some fresh perspectives are brought to the table in terms

of potential intervention ideas? How about having a mix of participants who were part of Work

Package 1 and some participants who were not?

Overall Feedback

Good proposal with the potential to garner some creative interventions that will be adapted by the

targeted communities.

Reviewer #2: The "PROCAN-B Study Protocol: Early Diagnosis of Prostate Cancer for Black Men - a community-centered Participatory Approach in Scotland and the North-East of England" aims to address the critical issue of health disparities in prostate cancer diagnosis among Black men through a community-centered, mixed methods research approach. While the protocol robustly outlines the involvement of the community via a Public Involvement and Community Engagement (PICE) group and integrates both qualitative and quantitative research methods, there are several areas where clarity and depth could be enhanced to improve the manuscript's strength and acceptability for publication. These areas include better integration of the mixed methods data, a more streamlined theoretical framework, detailed explanations of sample size justifications, and a more thorough discussion of ethical considerations and the intervention's specifics. Enhancing these aspects will provide clearer insights into the study's methodological rigor and potential impact. Additional comments below:

Based on the information provided from the sources, the PROCAN-B study protocol titled "The PROCAN-B Study Protocol: Early diagnosis of PROstate CANcer for Black men - a community-centered participatory approach in Scotland and the North-East of England" is a comprehensive approach designed to address disparities in prostate cancer diagnosis and treatment among Black men. The study employs a mixed methods design and emphasizes community participation and cultural tailoring of interventions, which are its significant strengths. However, for this study protocol to be enhanced and made more acceptable for publication in a peer-reviewed journal, the following areas could benefit from improvements or further clarity:

INTRODUCTION

• The introduction clearly states the problem and the target population, emphasizing the health disparity for prostate cancer in Black men in the UK. However, it could be enhanced by adding a concise review of the existing literature that highlights gaps the study aims to fill. Providing a brief overview of prior interventions and their limitations can strengthen the rationale for this study.

• Multiple theoretical frameworks are introduced (e.g., I-SAM, COM-B, socio-ecological model), but their interrelations and specific contributions to the research questions could be articulated more clearly to avoid conceptual overlap and confusion. I also think the whole protocol is too bloated with unnecessary use of frameworks without adequate justification of use.

• Additionally, articulating the theoretical framework that guides the study at the outset would lay a solid foundation for understanding the choice of methods and intervention design.

• While the socio-ecological conceptual framework is mentioned, its application to the study could be elaborated. How exactly does this framework guide the study's design and the development of interventions? A more detailed explanation would help in linking theoretical underpinnings directly to study actions and expected outcomes.

• Several abbreviations were not defined upon first use.

• Expanding on the rationale and benefits of using a participatory and community-centered approach for this specific population could solidify the study's methodological choice. Briefly discussing evidence that supports the effectiveness of such approaches in similar settings would strengthen the argument for adopting this strategy.

• There are mentions of the high incidence and mortality rates of prostate cancer among Black men, but we could benefit from more current or specific epidemiological data to underscore the urgency and scale of these disparities.

• The introduction asserts the importance of early diagnosis in closing health disparities but does not sufficiently address how early diagnosis can specifically mitigate these disparities. It could be beneficial to include evidence or theories suggesting that earlier diagnosis leads to better treatment outcomes and how these outcomes specifically impact the mortality and morbidity rates among Black men.

• There is mention of cultural and structural barriers; it would be strengthened by a deeper exploration of these factors. Discussing specific examples or past research findings that illustrate how these barriers manifest and affect health behaviors could provide more context and depth.

• The statement about the limited research in the UK on this topic is compelling but could be enhanced by a brief review of what has been done internationally and how this study's approach differs or builds on previous work.

METHODS

• The selection of study locations—Scotland and the North-East of England—raised questions about the demographic relevance, especially since areas like London and the West Midlands have a higher concentration of Black populations in the UK. The protocol mentions these regions are socioeconomically deprived with poor cancer outcomes, which might suggest a strategic focus on under-studied regions. However, for robustness and clarity, this choice warrants a more detailed justification.

• More specificity about the sampling strategy for focus groups, including how participants are identified and recruited, could enhance transparency and replicability. Detailing the inclusion and exclusion criteria would also fortify this section.

• The methods section should detail a specific integration strategy, such as a convergent design where both data types are collected simultaneously and analyzed separately, then merged to draw meta-inferences. This should include an explanation of how these inferences are synthesized to create comprehensive insights that neither method could achieve alone. Describing the analytical techniques, such as joint displays or narrative weaving, will further illustrate how data interrelate and support the study's conclusions.

• While the method for thematic analysis using NVivo is mentioned, the protocol could benefit from a more detailed description of the coding process, including how codes will be derived and how discrepancies between coders will be handled.

• The intervention's development is participatory, which is a strength. Yet, the description of how feedback from the community is systematically incorporated into the intervention development could be more explicit.

• The use of the Integrated Screening Action Model (I-SAM) is well-noted, but the manuscript could benefit from a more detailed explanation of how this model interacts with other theoretical frameworks mentioned, such as the COM-B model and the socio-ecological model. Clarifying how these models collectively inform the research questions, design, and expected outcomes could strengthen the theoretical basis.

o Also, while the I-SAM is intended to guide intervention development, specific examples of how its components (capability, opportunity, motivation) are operationalized within the study’s interventions would help in translating theory into practice more clearly.

• Each work package is aligned with specific objectives, but expanding on how these packages integrate with each other to achieve the overall study goals could clarify the step-by-step progression of the research.

o Also some sample sizes in the table do not match the descriptive texts.

• No detailed rationale for the chosen sample sizes in each work package. It's essential to clarify why specific numbers were selected and how they ensure statistical power or adequacy for meaningful analysis, especially considering the diversity and complexity of the population being studied.

• The protocol should specify whether the same participants are used across different WPs or if new participants are recruited for each phase.

o A clear flow diagram or a detailed description of participant progression through the study’s phases would aid in understanding how individuals move through the intervention, evaluation, and refinement stages

• More detail on the specific methods used in the evaluation phases (WP5 and WP8) and how findings from each phase feed into the next would underscore the iterative nature of the intervention development process.

• Pg 8: what does different ethnicities mean?

• How would findings from these areas be extrapolated or related to Black men in other UK regions?

• What exactly are the community engagement efforts that informed the choice of these locations. If local organizations or healthcare providers have existing relationships or have identified specific needs, these details would strengthen the rationale for site selection.

• Since this is a protocol, expected results should not be discussed, so it’s confusing why there are some results presented from some of the work packages.

DISCUSSION

• The protocol could pre-emptively address how it intends to interpret and disseminate the findings, especially in light of potential limitations. How does the study's design (including its participatory elements) impact the generalizability of the results to other settings or populations?

• What are the expected contributions to the literature, potential implications for practice and policy, and future research directions?

7. PLOS authors have the option to publish the peer review history of their article (what does this mean?). If published, this will include your full peer review and any attached files.

Reviewer #1: No

Reviewer #2: No

---

## [Author Response · Author response to Decision Letter 0]

4 Oct 2024

Many thanks for the constructive comments. A revision table has been added to demonstrate how we have responded to your comments.

---

## [Decision Letter · Decision Letter 1]

25 Nov 2024

The PROCAN-B Study Protocol: Early diagnosis of PROstate CANcer for Black men - a community-centred participatory approach in Scotland and the North-East of England

PONE-D-23-32907R1

Dear Dr. Christie-de Jong,

We’re pleased to inform you that your manuscript has been judged scientifically suitable for publication and will be formally accepted for publication once it meets all outstanding technical requirements.

Kind regards,

Mohammed Faruk

Academic Editor

PLOS ONE

Additional Editor Comments (optional):

The article is now fully accepted for publication

Reviewers' comments:

Reviewer's Responses to Questions

**Comments to the Author**

1. Does the manuscript provide a valid rationale for the proposed study, with clearly identified and justified research questions?

Reviewer #3: No

2. Is the protocol technically sound and planned in a manner that will lead to a meaningful outcome and allow testing the stated hypotheses?

Reviewer #3: Partly

3. Is the methodology feasible and described in sufficient detail to allow the work to be replicable?

Reviewer #3: Yes

4. Have the authors described where all data underlying the findings will be made available when the study is complete?

Reviewer #3: Yes

5. Is the manuscript presented in an intelligible fashion and written in standard English?

Reviewer #3: Yes

6. Review Comments to the Author

You may also provide optional suggestions and comments to authors that they might find helpful in planning their study.

Reviewer #3: While the revisions are extensive and address many of the prior concerns, several issues remain outstanding and warrant further attention:

1) Although the revised manuscript acknowledges the pilot nature of the study and aligns with typical sample sizes for similar research, it still lacks a detailed explanation of how the sample size ensures statistical power for meaningful analysis.

2) The manuscript references a sequential exploratory design and thematic analysis, but it does not adequately clarify how qualitative and quantitative findings will be integrated to provide comprehensive insights. The term "joined display" used in the revision should be corrected to "joint display." Additionally, the justification for employing a mixed methods approach remains unclear, and the study design does not convincingly demonstrate the characteristics of a true mixed methods study.

3) While the authors discuss the limited generalizability of the findings due to the study's design, they could provide a clearer articulation of how these results might inform broader applications or future research across other settings and populations.

7. PLOS authors have the option to publish the peer review history of their article (what does this mean?). If published, this will include your full peer review and any attached files.

Reviewer #3: No

---

## [Editor Report · Acceptance letter]

12 Dec 2024

PONE-D-23-32907R1 

PLOS ONE

Dear Dr. Christie-de Jong, 

I'm pleased to inform you that your manuscript has been deemed suitable for publication in PLOS ONE. Congratulations! Your manuscript is now being handed over to our production team.

Kind regards, 

on behalf of

Dr. Mohammed Faruk 

Academic Editor

PLOS ONE